**Brief Communication**

# Square beams for optimal tiling in transmission electron microscopy

Eugene Y. D. Chua [1,4], Lambertus M. Alink[1,4], Mykhailo Kopylov[1], Jake D. Johnston[1,2], Fabian Eisenstein [3] & Alex de Marco [1,2]✉

Imaging large fields of view at a high magnification requires tiling. Transmission electron microscopes typically have round beam profiles; therefore, tiling across a large area is either imperfect or results in uneven exposures, a problem for dose-sensitive samples. Here, we introduce a square electron beam that can easily be retrofitted in existing microscopes, and demonstrate its application, showing that it can tile nearly perfectly and deliver cryo-electron microscopy imaging with a resolution comparable to conventional set-ups.

In transmission electron microscopy (TEM) of dose-sensitive specimens such as vitrified biological material, pre-exposure of areas to the beam attenuates the attainable resolution[1]. High-resolution cryo-TEM typically comes at the cost of a reduced field of view; therefore, a balance between the pixel size and the sample imaged in its biological context is required. Given that the illumination profile (which we call here 'beam' or 'beam profile' for simplicity) of TEMs is round, tiling across a large field of view encounters the circle packing problem, wherein circles cannot be perfectly tiled. Even with an ideal modern imaging set-up with fringe-free imaging (FFI)[2,3] and a square sensor, the sensor will capture only ~69% of the area illuminated by the tightest possible round beam (Fig. 1 and Supplementary Fig. S1). The electron beam will damage the remaining illuminated but unimaged site, which will no longer contain high-resolution information when next imaged. This is a well-known limitation in montage tomography of vitrified specimens, and although data collection schemes that account for overlapping exposures exist[4,5], the illumination across multiple exposed areas remains non-uniform.

One solution to the problem of imperfect tiling with round beams is to use a square electron beam. Modern cryo-TEMs use Mueller-type sources in which the emitter's shape defines the electron beam shape, typically resulting in a circle. For TEM imaging in a 3-condenser (3-C) system, the beam current (spot size) is selected by the C1 and C2 lenses, and the source beam width (beam convergence) is determined by changing the strength of the C2 and C3 lenses. The aperture between the C2 and C3 lenses becomes the beam-shaping aperture. When the electron beam cross-over above the C2 aperture is moved (by changing the C1 and C2 lenses), the beam current changes, whereas when the cross-over below the C2 is moved (by changing the C2 and C3 lenses),

the size of the beam changes. The post-C2-aperture beam takes on the shape of the aperture's hole when the beam is spread wider than the aperture. For practical reasons linked to manufacturing and isotropic optical propagation, all apertures have round holes, creating round beams. In this work, we use a C2 aperture with a square hole to create a square electron beam profile. We demonstrate its utility on an FFI-capable TEM for near-perfect tiling in montage tomography and increased efficiency in data collection for single-particle analysis with minimal loss of resolution.

Using a square C2 aperture, we successfully created a square beam (Fig. 1c). First, we adjusted the beam width using the microscope intensity control such that the beam had the same size as the shortest dimension of the sensor. Then, the (post-objective) projection P2 lens was adjusted to rotate the beam square onto the sensor (Fig. 1c). Aligning the beam with the sensor ensures that the sensor images the entire sample area exposed to the beam. Given that changes in the P2 lens strength change the image's rotation, magnification and defocus, calibrations for pixel size, image shift and eucentric focus had to be redone. The flux on the sensor can be adjusted with spot size, and the beam intensity distribution across the illuminated area can be measured to ensure uniform exposures (Supplementary Fig. S2). The unique P2 lens state can be stored as a unique magnification entry and added as a separate registry key.

With a square beam, it became possible to tile with minimal overlap to exhaustively image a large field of view. This is especially important for in situ tomography, in which it is often helpful to image large contiguous areas of a specimen, such as a lamella, at high resolution. Acquisition targets can be set along the tilt axis to overlap minimally

[1]Simons Electron Microscopy Center, New York Structural Biology Center, New York, NY, USA. [2]Department of Physiology and Cellular Biophysics, Columbia University, New York, NY, USA. [3]Graduate School of Medicine, University of Tokyo, Tokyo, Japan. [4]These authors contributed equally: Eugene Y. D. Chua, Lambertus M. Alink. ✉e-mail: alex@nysbc.org

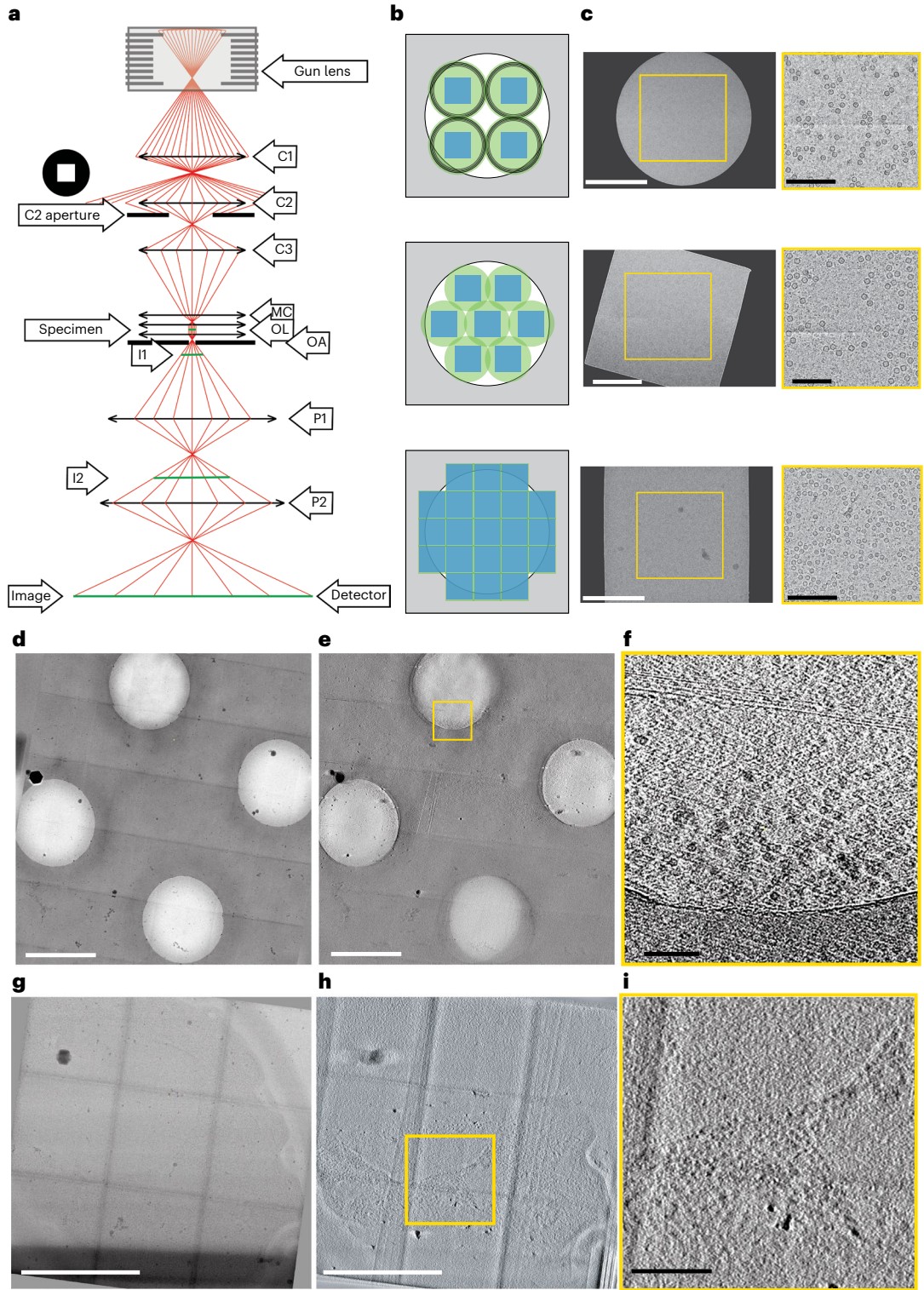

**Fig. 1 | Square beam set-up and montage tomography. a**, Ray diagram of electrons passing through the column of a TEM. The square aperture is placed at the C2 aperture position. C1–3, condenser lens 1–3; I1,2, intermediate image plane 1,2; MC, mini-condenser; OA, objective aperture; OL, objective lenses; P1,2, projection lens 1,2. **b**, Examples of imaging a specimen (white circle) with different TEM beam set-ups: top, imaging with a round beam; middle, FFI with a round beam; bottom, FFI with a square beam. The areas on the specimen illuminated by the electron beam are shown in green circles (round beam) or green squares (square beam); in non-FFI TEM set-ups the beam must be spread out so the fringes do not fall on the sensor. **c**, Example micrographs from an FFI-enabled TEM, acquired with a round beam (top), square beam (middle), and a square beam aligned to the sensor by adjusting the projection lens (bottom). **d**, Example of PACE-tomo tiled imaging with the square beam. We collected a 5 × 5 butt-joint tile set on holey carbon grids with apoferritin. **e**, The resulting tomogram reconstructed from the data shown in **d. f**, High-magnification crop of the joint between two tiles in **e**. In the upper section, although alignment or interpolation was not performed when stitching, the image shifts are sufficiently accurate to provide contextual information. Apoferritin particles are clearly visible even at the stitching lines in the reconstructed tomograms. **g**, PACE-tomo tiled imaging with the square beam on yeast lamellae. Here, we collected a 3 × 3 butt-joint tile set. **h**, The resulting tomogram reconstructed from the data in **g. i**, High-magnification crop of a region of the tomogram in **g**. Scale bars: **c** (left), 200 nm; (right) 100 nm; **d**,**e**, 1 μm; **f**,**i**, 100 nm; **g**,**h**, 500 nm.

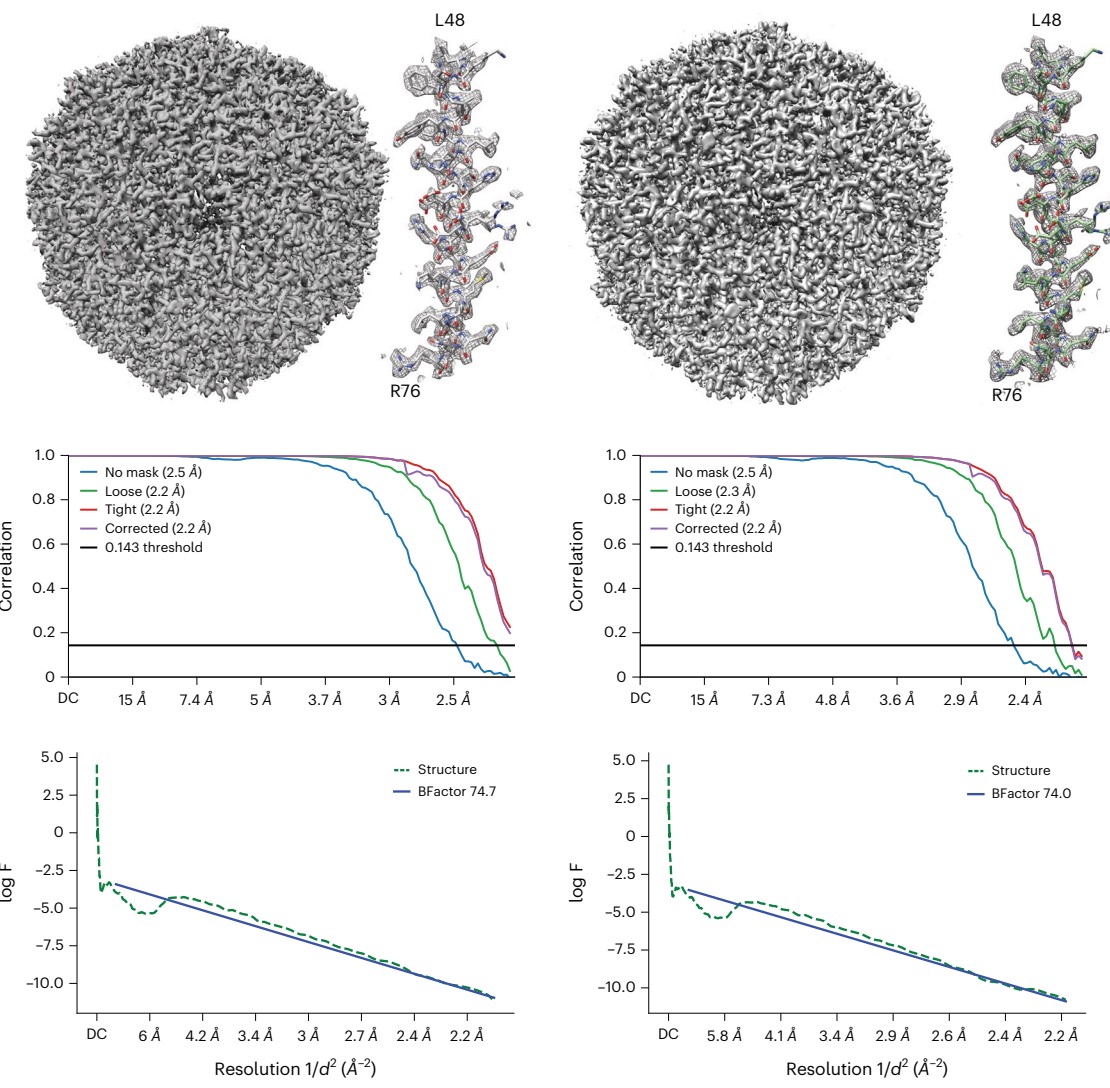

**Fig. 2 | Performance in single-particle analysis.** Single-particle reconstructions of apoferritin with data collected with a round (left) or a square (right) beam. In both cases, with 120,000 particles, the reconstructions can achieve Nyquist resolution. The middle graphs show the gold-standard Fourier shell correlation (GSFSC) curves (with an overall resolution of 2.16 Å in both cases) and the lower graphs show the Guinier plots (d, the resolution (Å); F, the spherically averaged structure factor amplitude).

and, therefore, to reduce any areas on the sample that are exposed to the electron beam in more than one acquisition target. To maximize the acquisition area while avoiding the sample overexposure in the direction perpendicular to the tilt axis, a new data acquisition scheme was developed, in which the beam-image shift is independent of the sample tilt. This scheme was implemented using PACE-tomo (parallel cryo-electron tomography)[6], and the beam shifts were set to be equivalent to the length of the y-dimension of the high-magnification image, in nm (Fig. 1d–i and Supplementary Videos 1–4). The beam overlap was maintained identically throughout the tilt series (Supplementary Fig. S3 shows the difference between conventional PACE-tomo tiled tracking versus camera-based offset). After data acquisition, a montage for each stage tilt can be stitched to produce a sizeable field-of-view image, which is then aligned across the tilt series and reconstructed. Although this data collection method eliminates sample overexposure, the lack of overlapping regions between tiles results in visible stitching lines (for example, Fig. 1d,g). We tested an alternate data collection scheme of a 3 × 3 montage on a carbon-foil apoferritin grid in which each tile had a 5% overlap with its neighbor. The overlapping regions were then cross-correlated and blended with each other[4,5] to produce a nearly seamless montage (Supplementary Fig. S4).

In single-particle data acquisition, a square beam significantly increases throughput. Using common supports such as UltrAuFoil R1.2/1.3 grids, a square beam could image up to five targets per hole (85 targets per stage movement) versus two targets per hole for a round beam (Supplementary Fig. S5). This increased the data collection rate, from ~158 exposures per hour with the round beam to ~291 exposures per hour with the square beam.

During the microscope set up we observed normal behavior during microscope alignment and coma correction (Supplementary Fig. S6), and with enough particles, reconstructions from round and square beams both achieved Nyquist resolution (Fig. 2). With smaller particle sets, we consistently observed a slight loss in reconstruction resolution (~0.1 Å) and a lower reconstruction B-factor with the square beam than the round beam (Supplementary Fig. S7 and Supplementary Table S1). This is most likely to be linked to the lack of circular symmetry in the phase profile of the beam diffraction from the square aperture. A potential solution is to use larger apertures to lower the diffraction angle and maintain a more uniform phase profile at the sample plane.

The most notable change from detuning the P2 lens is the change in pixel size. After obtaining a high-resolution map of apoferritin, a model was fitted to the map, and the pixel size was recalibrated by

optimizing the map's fitting to the model. When we detuned our P2 lens, we observed a change in pixel size from 1.063 to 1.038 Å per pixel. The Cs (spherical aberration) of the microscope did not change.

Considering the change in the tuning of the P2 lens, we tested the potential changes in microscope performance. The magnification was not subject to any measurable distortion (Supplementary Table S2)[7]. Further, Young's fringes experiment showed that the microscope performance remained within the manufacturer's specification as the information transmittance reached 0.14 nm on gold cross-grating (Supplementary Fig. S8).

Non-circular and square beams have been developed for beam-shaping in electron beam lithography, laser micromachining and medical laser applications. We use a square beam for nearly perfect cryo-electron microscopy and cryo-electron tomography montage tiling. Other non-round beams, such as rectangles and hexagons, which are also optimal profiles for tiling, can be used[8]. We note that it is possible to create a rectangular beam by stigmating a square beam; however, this introduces unwanted aberrations into the beam. Furthermore, matching the beam and detector's shapes prevents minor data processing complications associated with having unilluminated sensor areas.

With regard to optimization, the alignment of the aperture orientation with respect to the sensor is critical to ensure optimal overlap between the sensor and the illuminated area. As an alternative to rotating the beam with the P2 lens, we envision that the aperture can be mechanically rotated through a redesign of the aperture strip: a worm wheel gear can be installed to physically rotate the square aperture while it is in the liner tube under vacuum during illumination. For this work, we propose a no-cost solution that involves adjusting the P2 lens current to induce a rotation of the projected image plane. This introduces changes in the optical system, affecting the image's magnification, defocus, and rotation, requiring re-calibration of the pixel size, eucentric focus and image shift matrices.

## Online content

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

## Methods

Square apertures were purchased from Agar Scientific (product numbers AGAS3005P and AGAS3010P). The apertures are made of platinum, with a diameter of 3.04 mm, a thickness of 0.25 mm and a square hole of 50 or 100 μm. A 50 μm aperture has been installed into the C2 aperture holder of the microscope. Before its installation into the microscope, the square aperture should be plasma cleaned to remove any impurities, then maintained in a sealed container for several days to enable the charge to dissipate and facilitate an easier insertion into the aperture strip.

The optics of the electron microscope consist of three sections: the beam-forming condenser section where the square aperture is installed, the objective section with the specimen, and the magnifying projection lens section (Fig. 1a). The square aperture is located at the condenser section, defining the square beam. To align the square beam towards the camera, we tuned the P2 projection lens. The condenser and objective lens settings were not touched to keep the parallelism of the beam intact going through the objective system. With the electron microscope set to diffraction mode, on detuning of the condenser lens, the diffraction rings showed blurring, indicating loss of parallelism of the beam in the objective lens. Detuning of the projection system left the parallelism intact. This was expected given that the projection system is below the objective lens. Note that tuning the projection lens will affect the image's magnification, rotation and defocus.

The strength of the P2 lens can be checked in the user interface system status overview (Supplementary Fig. S9). The square aperture can be inserted using the standard OCX software in the user interface. The square C2 aperture has the same form factor as the 'standard' round C2 aperture. Replacement of the C2 aperture, aligning the aperture laterally, and tuning the P2 lens can be done by the equipment supplier service engineer using the standard software and wizards. A screenshot of the aperture wizard is given in Supplementary Fig. S10. After the adjustments, eucentric focus calibration, pixel size and image shift calibrations must be performed using the SerialEM program[9].

The same protein sample was used for apoferritin tomography and single-particle analysis prepared as described below, with the only difference found in the support (carbon versus gold). Data were acquired using PACE-tomo scripts[6] incorporated into SerialEM 4.1 beta 13 (ref. 9) with a pixel size of 2.12 Å per pixel, exposure dose of 3.4 e⁻ Å⁻² per tilt, a tilt range of −45° to 45° for a total of 31 tilts and a total dose of 105 e⁻ Å⁻² per tilt series. Imaging was done as 5 × 5 patches. A *Saccharomyces cerevisiae* standard yeast test sample for lamella tomography was prepared following the Waffle method protocol[10,11]. Data were acquired using PACE-tomo scripts incorporated into SerialEM 4.1 beta 13 with a pixel size of 2.12 Å per pixel, a tilt exposure dose of 2.55 e⁻ Å⁻² and a total dose of 76.5 e⁻ Å⁻². Imaging was done as 3 × 3 patches to encompass the whole yeast cell in the field of view.

For the apoferritin dataset, the acquired tilt series were first motion corrected with Warp[12], and tilt series were aligned and reconstructed with AreTomo[13] at bin6 and used without additional processing for further analysis. The tilt series were motion corrected with Warp 12 for the yeast lamella dataset and aligned using AreTomo[13]. Tomogram reconstruction was performed using Tomo3D (ref. 14), and the tomogram was deconvolved with IsoNet[15] to enhance contrast. Stitching for both the apoferritin and yeast lamella datasets was done automatically with custom Python scripts, in which the tiles were stitched together using the image shift locations obtained from the metadata.

For single-particle analysis, UltrAuFoil R1.2/1.3 300 mesh Au grids were hydrophilized with a mixture of Ar and O₂ gas (26.3:8.7 ratio) at 15 W for 7 s in a Solarus Model 950 Advanced Plasma System (Gatan). A total of 3 μl of 8 mg ml⁻¹ mouse apoferritin was pipetted onto each grid, blotted for 3–5 s in a Vitrobot at 20 °C and 100% relative humidity, then vitrified in liquid ethane. The P2 projection lens was detuned to rotate the square beam square onto the sensor, resulting in changes in the image's magnification, rotation and defocus. Eucentric focus

needed to be reset on the microscope by adjusting the objective lens, then beam and image shift and scale rotation calibrations were required to be redone in Leginon[16,17] prior to data collection. Pixel size calibration was done in SerialEM on a standard cross-grating replica grid. Energy filter alignments were done per standard protocol, with the entire sensor illuminated. Objective lens astigmatism and coma correction were performed using Sherpa, with the full sensor illuminated. Single-particle data were collected using Leginon with either a 100 μm round C2 aperture or a 50 μm square C2 aperture. Data were collected at a pixel size of ~1.08 Å per pixel, a flux of ~30 e⁻ px⁻¹ s⁻¹ for 2 s, equaling a total dose of ~51 e⁻ Å⁻², with a nominal defocus range of −0.5 μm to −2.0 μm. The square beam was condensed to match the size of the sensor to maximize the data acquisition area, and in the control experiment a round beam was used with its intensity set to match the flux of the square beam on the sensor. Data were collected using beam-image shift[18].

For single-particle data processing, square and round beam data were first motion corrected with patches in cryoSPARC v4.2.1 (ref. 19). Full micrographs then underwent patch CTF (contrast transfer function) estimation, and particles were picked using apoferritin templates. Particles were extracted with a box size of 280 pixels, 2D classified, and 120,000 particles randomly selected for homogeneous refinement. To exclude the unilluminated areas of the sensor because of the condensed square beam, a central square region of the motion-corrected micrographs was cropped out with the IMOD[20] command 'trimvol'. An identical central square region was cropped from both square and round aperture data as a control. Cropped micrographs were then re-imported into cryoSPARC, and the same processing workflow continued as above. Reconstructions from full and cropped micrographs from square and round aperture data were compared.

### Reporting summary

Further information on research design is available in the Nature Portfolio Reporting Summary linked to this article.

## Data availability

Single-particle analysis movies of apoferritin with and without P2 lens rotation and with square or round apertures have been deposited in the Electron Microscopy Public Image Archive (EMPIAR) with the accession code EMPIAR-11731. Accompanying apoferritin reconstructions have been deposited in the Electron Microscopy Data Bank (EMDB) with the accession codes EMD-42371, EMD-42372, EMD-42373 and EMD-42374. Tilted montage movies of apoferritin on a carbon foil grid have been deposited in EMPIAR with the accession code EMPIAR-11771. The accompanying tomogram has been deposited in EMDB with the accession code EMD-42851. Tilted montage movies of yeast lamella have been deposited in EMPIAR with the accession code EMPIAR-11778. The accompanying tomogram has been deposited in EMDB with the accession code EMD-42879.

## Code availability

The PACE-tomo code is available at https://github.com/eisfabian/PACEtomo.

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

## Acknowledgements

The authors thank M. Kikkawa (University of Tokyo) for the apoferritin plasmid and B. Kloss (NYSBC) for the expression and purification of the protein. The authors also thank A. Cheng, B. Carragher and C. Potter (Chan Zuckerberg Imaging Institute) for their early discussions and support, and S. Katznelson (ThermoFisher Scientific) for his engineering support. This work was supported by the Simons Electron Microscopy Center and National Resource for Automated Molecular Microscopy located at the New York Structural Biology Center, supported by grants from the Simons Foundation (SF349247) and the NIH National Institute of General Medical Sciences (GM103310).

## Author contributions

E.Y.D.C.: conceptualization, investigation, project administration, writing – original draft, writing – review and editing. L.M.A.: conceptualization, methodology, investigation, writing – review and editing. M.K.: investigation, writing – review and editing. J.D.J.: formal analysis, writing – review and editing. F.E.: software. A.d.M.: writing – review and editing, funding acquisition.

## Competing interests

The authors declare no competing interests.

## Additional information

**Correspondence and requests for materials** should be addressed to Alex de Marco.

# Reporting Summary

## Statistics

For all statistical analyses, confirm that the following items are present in the figure legend, table legend, main text, or Methods section.

| n/a | Confirmed | |
|---|---|---|
| ☐ | ☒ | The exact sample size (*n*) for each experimental group/condition, given as a discrete number and unit of measurement |
| ☐ | ☒ | A statement on whether measurements were taken from distinct samples or whether the same sample was measured repeatedly |
| ☒ | ☐ | The statistical test(s) used AND whether they are one- or two-sided *Only common tests should be described solely by name; describe more complex techniques in the Methods section.* |
| ☒ | ☐ | A description of all covariates tested |
| ☒ | ☐ | A description of any assumptions or corrections, such as tests of normality and adjustment for multiple comparisons |
| ☒ | ☐ | A full description of the statistical parameters including central tendency (e.g. means) or other basic estimates (e.g. regression coefficient) AND variation (e.g. standard deviation) or associated estimates of uncertainty (e.g. confidence intervals) |
| ☒ | ☐ | For null hypothesis testing, the test statistic (e.g. *F*, *t*, *r*) with confidence intervals, effect sizes, degrees of freedom and *P* value noted *Give P values as exact values whenever suitable.* |
| ☒ | ☐ | For Bayesian analysis, information on the choice of priors and Markov chain Monte Carlo settings |
| ☒ | ☐ | For hierarchical and complex designs, identification of the appropriate level for tests and full reporting of outcomes |
| ☒ | ☐ | Estimates of effect sizes (e.g. Cohen's *d*, Pearson's *r*), indicating how they were calculated |

*Our web collection on statistics for biologists contains articles on many of the points above.*

## Software and code

Policy information about availability of computer code

| Data collection | Data were collected using SerialEM, PACE-tomo (see Code availability), and Leginon on a ThermoFisher Titan Krios G2 |
|---|---|
| Data analysis | Data were analized using CryoSPARC, IMOD, Aretomo, Warp, Isonet, Tomo3d |

For manuscripts utilizing custom algorithms or software that are central to the research but not yet described in published literature, software must be made available to editors and reviewers. We strongly encourage code deposition in a community repository (e.g. GitHub). See the Nature Portfolio guidelines for submitting code & software for further information.

## Data

Policy information about availability of data

All manuscripts must include a data availability statement. This statement should provide the following information, where applicable:
- Accession codes, unique identifiers, or web links for publicly available datasets
- A description of any restrictions on data availability
- For clinical datasets or third party data, please ensure that the statement adheres to our policy

Single particle analysis movies of apoferritin with and without P2 lens rotation and with square or round apertures have been deposited in EMPIAR with the accession code EMPIAR-11731. Accompanying apoferritin reconstructions have been deposited in EMDB with the accession codes EMD-42371, EMD-42372, EMD-42373, and EMD-42374.

Tilted montage movies of apoferritin on a carbon foil grid have been deposited in EMPIAR with the accession code EMPIAR-11771. The accompanying tomogram has been deposited in EMDB with the accession code EMD-42851.

Tilted montage movies of yeast lamella have been deposited in EMPIAR with the accession code EMPIAR-11778. The accompanying tomogram has been deposited in EMDB with the accession code EMD-42879.

## Human research participants

Policy information about studies involving human research participants and Sex and Gender in Research.

| | |
|---|---|
| Reporting on sex and gender | N/A |
| Population characteristics | N/A |
| Recruitment | N/A |
| Ethics oversight | N/A |

Note that full information on the approval of the study protocol must also be provided in the manuscript.

# Field-specific reporting

Please select the one below that is the best fit for your research. If you are not sure, read the appropriate sections before making your selection.

☒ Life sciences        ☐ Behavioural & social sciences        ☐ Ecological, evolutionary & environmental sciences

For a reference copy of the document with all sections, see nature.com/documents/nr-reporting-summary-flat.pdf

# Life sciences study design

All studies must disclose on these points even when the disclosure is negative.

| | |
|---|---|
| Sample size | Both single particle dataset consist of 120000 particles. The sample size has been determined through picking using cryoSPARC with manual curation of the process to ensure correct performance. |
| Data exclusions | No data have been excluded |
| Replication | the reconstructions have been validated through independent reconstructions following the current best practice in the field. All measurements, and system calibrations have been performed a minimum of 2 times independently everytime the microscope was updated (configuration, magnification) |
| Randomization | N/A, structural determination in Single particle does not require randomisation as it is the result of averaging of a large pool of individual projection images of proteins. Picking of the projections is performed automatically and curation is only done with the purpose of controlling the uniformity of the output. All the measures were performed to quantify the changes (if any) induced by the modifications to the microscope configuration. Randomisation was not required as no difference was appreciated in the final resolution. |
| Blinding | N/A, Blinding was not required as no difference was appreciated in performance when modifying the aperture profile. |

# Reporting for specific materials, systems and methods

We require information from authors about some types of materials, experimental systems and methods used in many studies. Here, indicate whether each material, system or method listed is relevant to your study. If you are not sure if a list item applies to your research, read the appropriate section before selecting a response.

### Materials & experimental systems

| n/a | Involved in the study |
|---|---|
| ☒ ☐ | Antibodies |
| ☒ ☐ | Eukaryotic cell lines |
| ☒ ☐ | Palaeontology and archaeology |
| ☒ ☐ | Animals and other organisms |
| ☒ ☐ | Clinical data |
| ☒ ☐ | Dual use research of concern |

### Methods

| n/a | Involved in the study |
|---|---|
| ☒ ☐ | ChIP-seq |
| ☒ ☐ | Flow cytometry |
| ☒ ☐ | MRI-based neuroimaging |

