## [Peer Review File · Nature Methods]

Peer Review Information

Manuscript Title: Square beams for optimal tiling in TEM

Corresponding author name(s): Alex de Marco

Editorial Notes: None

Reviewer Comments & Decisions:

Decision Letter, initial version:

Dear Alex,

Your Brief Communication, "Square beams for optimal tiling in TEM", has now been seen by three reviewers. As you will see from their comments below, although the reviewers find your work of considerable potential interest, they have raised a number of concerns. We are interested in the possibility of publishing your paper in Nature Methods, but would like to consider your response to these concerns before we reach a final decision on publication.

We therefore invite you to revise your manuscript to address these concerns. We ask that you focus your efforts on clarifying concerns, creating a streamlined workflow for implementing the idea in practice, and demonstrating on a real-world challenging montage application. During revision, you may expand your paper to an Article if length and number of figures is a concern.

* include a point-by-point response to the reviewers and to any editorial suggestions

* please underline/highlight any additions to the text or areas with other significant changes to facilitate review of the revised manuscript

- * address the points listed described below to conform to our open science requirements
- * ensure it complies with our general format requirements as set out in our guide to authors at www.nature.com/naturemethods
- * resubmit all the necessary files electronically by using the link below to access your home page

[Redacted] This URL links to your confidential home page and associated information about manuscripts you may have submitted, or that you are reviewing for us. If you wish to forward this email to co-authors, please delete the link to your homepage.

We hope to receive your revised paper within three months. If you cannot send it within this time, please let us know. In this event, we will still be happy to reconsider your paper at a later date so long as nothing similar has been accepted for publication at Nature Methods or published elsewhere.

OPEN SCIENCE REQUIREMENTS

REPORTING SUMMARY AND EDITORIAL POLICY CHECKLISTS

Please note that these forms are dynamic 'smart pdfs' and must therefore be downloaded and completed in Adobe Reader. We will then flatten them for ease of use by the reviewers. If you would

like to reference the guidance text as you complete the template, please access these flattened versions at <http://www.nature.com/authors/policies/availability.html>.

DATA AVAILABILITY

We strongly encourage you to deposit all new data associated with the paper in a persistent repository where they can be freely and enduringly accessed. We recommend submitting the data to discipline-specific and community-recognized repositories; a list of repositories is provided here:

<http://www.nature.com/sdata/policies/repositories>

All novel DNA and RNA sequencing data, protein sequences, genetic polymorphisms, linked genotype and phenotype data, gene expression data, macromolecular structures, and proteomics data must be deposited in a publicly accessible database, and accession codes and associated hyperlinks must be provided in the “Data Availability” section.

Please include a “Data availability” subsection in the Online Methods. This section should inform readers about the availability of the data used to support the conclusions of your study, including accession codes to public repositories, references to source data that may be published alongside the paper, unique identifiers such as URLs to data repository entries, or data set DOIs, and any other statement about data availability. At a minimum, you should include the following statement: “The data that support the findings of this study are available from the corresponding author upon request”, describing which data is available upon request and mentioning any restrictions on availability. If DOIs are provided, please include these in the Reference list (authors, title, publisher (repository name), identifier, year). For more guidance on how to write this section please see: <http://www.nature.com/authors/policies/data/data-availability-statements-data-citations.pdf>

CODE AVAILABILITY

Please include a “Code Availability” subsection in the Online Methods which details how your custom code is made available. Only in rare cases (where code is not central to the main conclusions of the paper) is the statement “available upon request” allowed (and reasons should be specified).

For more information on our code sharing policy and requirements, please see:
<https://www.nature.com/nature-research/editorial-policies/reporting-standards#availability-of-computer-code>

MATERIALS AVAILABILITY

ORCID

Nature Methods is committed to improving transparency in authorship. As part of our efforts in this direction, we are now requesting that all authors identified as ‘corresponding author’ on published papers create and link their Open Researcher and Contributor Identifier (ORCID) with their account on the Manuscript Tracking System (MTS), prior to acceptance. This applies to primary research papers only. ORCID helps the scientific community achieve unambiguous attribution of all scholarly contributions. You can create and link your ORCID from the home page of the MTS by clicking on ‘Modify my Springer Nature account’. For more information please visit www.springernature.com/orcid.

Sincerely,
Rita

Rita Strack, Ph.D.
Senior Editor
Nature Methods

Reviewers' Comments:

Reviewer #1:

Remarks to the Author:

This is an interesting paper, and the tiling method is good for collecting lamella data since we always want to image the whole lamella. However, the installation of the square-shaped aperture may not be that convenient? Overall alignment procedures may complicate and beam is not isotropic in all directions compare to a round beam. It is necessary to conduct an anisotropic magnification distortion test on the image in comparison to an image captured with a round beam, aiming to identify any potential alterations.

One of the question is what is strategy in your experiment drive the decision of the size of the square aperture and how to adjust the square size? And why not use the rectangular aperture which can maximize the usage of the camera detector and unilluminated area of sensor problem in motion correction?

Also, even with a square beam, the overlap is still necessary if we want a seamless montage. This is a trade-off between having less area being beam damaged twice and going through the process of testing and calibrating the square aperture. But overall this idea is great, if this whole protocol can be further developed (to minimize the tile overlap, make sure the stability of the beam rotation, compatibility with different cameras, and develop algorithms to minimize extra exposure when tilting), it will become a very useful method.

Here is a minor comment:

In Supplementary Figure 3, it will be clearer also include both the beam area and detector area.

Reviewer #2:

Remarks to the Author:

A. Summary of the key results:

Montage tomography is a useful technique for capturing a comprehensive cellular context of macromolecules of interest with high resolution. By allowing the imaging of a large field of view while maintaining high-magnification data collection, it offers an opportunity to study the cellular details within a broader context.

The method relies on repeated overlapping exposures to cover the entire area of interest, but this practice introduces a concern: the adjoining tiles, where the exposures overlap, are susceptible to sample damage due to cumulative radiation exposure.

Addressing this issue, two recent papers have presented strategies in mitigating the potential damage. The first paper by Peck et al. (2022) titled "Montage electron tomography of vitrified specimens", which introduces a clever strategy of applying a global translational image shift. This shift not only optimizes the distribution of additional radiation dose throughout the specimen but also effectively minimizes the impact of overlapping beam exposures.

A complementary advancement comes from the work of Yang et al. (2023) in their paper "Correlative cryogenic montage electron tomography for comprehensive in-situ whole-cell structural studies," where they further enhance the technique.

To address the challenge of the circular beam overlapping a rectangular or square detector, the authors of this paper adopt a square C2 aperture. By generating a beam with a matching square shape, they align the geometries of the beam and detector, eliminating the issue of uneven overlap and potential damage at the stitched areas.

B. Originality and significance: if not novel, please include referenceB.

While both the global translational image shift and the use of the square aperture work synergistically to optimize the over-exposure damage of cryo-sample, the novel method presented in this work not only addresses the technical hurdles that have hindered the full potential of montage cryo-electron tomography but also broadens the scope for comprehensive in-situ cellular studies.

C. & D. Data & methodology: validity of approach, quality of data, quality of presentation

Appropriate use of statistics and treatment of uncertainties

This paper demonstrated a good tiling montage along the stage tilt axis with very minimal overlapping exposure. While the situation changes when tiling occurs in the direction perpendicular to the tilt axis. During stage tilting, adjacent tiles unavoidably experience overlapping exposure, leading to radiation damage. Therefore, a new data acquisition scheme which considers the image shift relative to the stage tilt angle is required to balance the over-exposure.

In this study, the authors undertook a crucial step to optimize their imaging setup. They focused on adjusting the P2 projection lens to ensure that the square illuminated area was precisely aligned with the orientation of the detector. While this adjustment guarantees imaging quality, changing of P2

requires a series of re-calibration of pixel size, focus, image shift, etc., which is not optimal and efficient for routine data collection setup. The authors put forward an alternative solution that mechanically align the square C2 aperture with the detector orientation by installing a worm wheel gear on the aperture strip.

E. Conclusions: robustness, validity, reliability

In conclusion, while the paper contributes to the emerging trend of montage tomography by introducing an innovative methodology. The limited amount of preliminary data presented implies that further experimentation, validation is necessary to fully assess the effectiveness and robustness of the method.

F. Suggested improvements: experiments, data for possible revision

To establish the application a square beam for montage tomography, it is advisable to supplement the paper's findings with successful montaged tomograms obtained through larger tiling configurations, such as 3x3 or even larger. Especially, the over-exposure issue at adjoining tiles perpendicular to tilt axis need to be addressed.

G. References: appropriate credit to previous work?

This work builds on previous research in the field. The authors have cited all the important findings in the references.

H. Clarity and context: lucidity of abstract/summary, appropriateness of abstract, introduction and conclusions

The abstract and text are all clear.

Reviewer: Xiaowei Zhao

Reviewer #3:

Remarks to the Author:

A. The authors employed a square beam profile combined with fringe free illumination to aim for near-perfect tiling in montage cryo electron tomography.

B. Since detector-matching square beam significantly reduces beam overlapping, a major benefit out of this work would be potentially, a high-resolution, 3-dimensional, and large field of view of, for example, cellular contexts.

Although the authors have shown tiled images at 0 and +/- 45 degrees, no montage tilt series is presented. In addition, the text at line 67 suggests that manual identification and alignment of

overlapping areas was performed. How long does this take? I understand this work is a snapshot of an early development, but is there any thought of automated image tiling over the entire tilt range? I do not think anyone would like to manually tile images together.

Line 28-30: “Cryo-TEM imaging requires balancing the entire region of interest.” This is true if we want to have both high resolution and large field of view. For other purposes like identifying holes of proper ice thickness and targets for data collection, montage are usually made at much lower magnifications. In these cases, we do not need high resolution and thus the balance.

Fig. 1C: Why do the square-beam images have much lower contrast than the round-beam counterpart? They are all FFI images except different beam shapes. The poor contrasts can also be seen in Supplemental Fig 2 at lines 264 and 268, respectively. Such differences do need explanations.

Fig 1D-E (1): The top and bottom edges of tile images have irregular shapes that change from tile to tile. If the irregular shapes correspond to the square C2 aperture, I do not understand why the shapes of tile images change from tile to tile. Please explain.

Fig 1D-E (2): There are also visible gaps (white strips) both in the line and square montages. I do not understand why the authors allow the presence of the gaps in their montage tilt series. Are the authors going to reconstruct 3D volumes from the montage tilt series? If so, what is the strategy to fill the tiling gaps? If not, please explain how the montage tilt series is used.

While square beam has a great potential to improve montage tilt series, the manuscript has not yet be convincing regarding how to efficiently and accurately tile images together and produce high-resolution large field view from the tiled tilt series. I would like to read the revised manuscript should the authors be willing to address these questions.

Author Rebuttal to Initial comments

Response to the reviewers

We thank the reviewers for the effort in evaluating this work and for the comments we believe improve the manuscript for the wider audience. Below we provide a point-to-point response to the comments in blue.

Reviewer #1:

Remarks to the Author:

This is an interesting paper, and the tiling method is good for collecting lamella data since we always want to image the whole lamella. However, the installation of the square-shaped aperture may not be that convenient? Overall alignment procedures may complicate and beam is not isotropic in all directions compare to a round beam. It is necessary to conduct an anisotropic magnification distortion test on the image in comparison to an image captured with a round beam, aiming to identify any potential alterations.

The installation of the square aperture can currently be performed by every service engineer in the field without additional training, and we suggest this should be performed through the manufacturers to ensure microscope warranty or service coverage is not hindered. The tuning of the projection system to match the orientation of the beam and the detector is also a task that a service engineer can perform, and the resulting new magnification setting can be saved in the registry so users do not need to perform any alignment.

As suggested by the reviewer, we performed a magnification distortion test to ensure no changes occurred on this front. We included the data in the supplementary table S1 and discussed it in the main text. The aberrations generally present become visible in the lower SA range and are dependent on the lens configuration. The tuning we perform to optimize the rotation is small enough that it does not change the range in which the lens operates. Accordingly, the distortion remains within the value for the equivalent magnification with a round beam.

We further analysed the performance of the microscope and show the result of the Young's fringes test in the supplementary figure S7. This test is part of the regular standard acceptance test performed during microscope installation.

One of the question is what is strategy in your experiment drive the decision of the size of the square aperture and how to adjust the square size? And why not use the rectangular aperture which can maximize the usage of the camera detector and unilluminated area of sensor problem in motion correction?

The initial tests were performed on an aperture that was available off-the-shelf. The use of rectangular apertures is expected to work identically to the square. The difference in diffraction angle between the two axes of the aperture is going to be negligible (0.044 urad vs 0.024 urad for the extreme case of a 50x100 um aperture). Also, rectangular apertures have been shown to work by Brown et al. BioRxiv 2023. We have added a citation to Brown et al. and mentioned the possibility of using other shapes than square to better match the detectors.

In terms of cropping, this can be done automatically, and the position of the illuminated area can be segmented based on the intensity in the image. Since the beam's position does not change between frames, only their content, this would be a possible preprocessing step to be added to the frame alignment.

Also, even with a square beam, the overlap is still necessary if we want a seamless montage. This is a trade-off between having less area being beam damaged twice and testing and calibrating the square aperture. But overall this idea is great; if this whole protocol can be further developed (to minimize the tile overlap, make sure the stability of the beam rotation, compatibility with different cameras, and develop algorithms to minimize extra exposure when tilting), it will become a very useful method.

We have expanded on the protocol to ensure full reproducibility in the supplementary material.

Here is a minor comment:

In Supplementary Figure 3, it will be clearer also include both the beam area and detector area. We modified the figure to include the reviewer suggestion.

Reviewer #2:

Remarks to the Author:

A. Summary of the key results:

Montage tomography is a useful technique for capturing a comprehensive cellular context of macromolecules of interest with high resolution. By allowing the imaging of a large field of view while maintaining high-magnification data collection, it offers an opportunity to study the cellular details within a broader context.

The method relies on repeated overlapping exposures to cover the entire area of interest, but this practice introduces a concern: the adjoining tiles, where the exposures overlap, are susceptible to sample damage due to cumulative radiation exposure.

Addressing this issue, two recent papers have presented strategies in mitigating the potential damage. The first paper by Peck et al. (2022) titled "Montage electron tomography of vitrified specimens", which introduces a clever strategy of applying a global translational image shift. This shift not only optimize the distribution of additional radiation dose throughout the specimen but also effectively minimizes the impact of overlapping beam exposures.

A complementary advancement comes from the work of Yang et al. (2023) in their paper "Correlative cryogenic montage electron tomography for comprehensive in-situ whole-cell structural studies," where they further enhance the technique.

To address the challenge of the circular beam overlapping a rectangular or square detector, the authors of this paper adopt a square C2 aperture. By generating a beam with a matching square shape, they align the geometries of the beam and detector, eliminating the issue of uneven overlap and potential damage at the stitched areas.

B. Originality and significance: if not novel, please include reference.

While both the global translational image shift and the use of the square aperture work synergistically to optimize the over-exposure damage of cryo-sample, the novel method presented in this work not only addresses the technical hurdles that have hindered the full potential of montage cryo-electron tomography but also broadens the scope for comprehensive in-situ cellular studies.

C. & D. Data & methodology: validity of approach, quality of data, quality of presentation
Appropriate use of statistics and treatment of uncertainties

This paper demonstrated a good tiling montage along the stage tilt axis with very minimal overlapping exposure. While the situation changes when tiling occurs in the direction perpendicular to the tilt axis. During stage tilting, adjacent tiles unavoidably experience

overlapping exposure, leading to radiation damage. Therefore, a new data acquisition scheme which considers the image shift relative to the stage tilt angle is required to balance the over-exposure.

In this study, the authors undertook a crucial step to optimize their imaging setup. They focused on adjusting the P2 projection lens to ensure that the square illuminated area was precisely aligned with the orientation of the detector. While this adjustment guarantees imaging quality, changing of P2 requires a series of re-calibration of pixel size, focus, image shift, etc., which is not optimal and efficient for routine data collection setup. The authors put forward an alternative solution that mechanically align the square C2 aperture with the detector orientation by installing a worm wheel gear on the aperture strip.

E. Conclusions: robustness, validity, reliability

In conclusion, while the paper contributes to the emerging trend of montage tomography by introducing an innovative methodology. The limited amount of preliminary data presented implies that further experimentation, validation is necessary to fully assess the effectiveness and robustness of the method.

F.Suggested improvements: experiments, data for possible revision

To establish the application a square beam for montage tomography, it is advisable to supplement the paper's findings with successful montaged tomograms obtained through larger tiling configurations, such as 3x3 or even larger. Especially, the over-exposure issue at adjoining tiles perpendicular to tilt axis need to be addressed.

We have added new data in Figure 1 as well as supplementary videos. Further we have adapted PACE-tomography to enable a simple implementation of tiled tomography.

G.References: appropriate credit to previous work?

This work builds on previous research in the field. The authors have cited all the important findings in the references.

H. Clarity and context: lucidity of abstract/summary, appropriateness of abstract, introduction and conclusions

The abstract and text are all clear.

Reviewer: Xiaowei Zhao

Reviewer #3:

Remarks to the Author:

A. The authors employed a square beam profile combined with fringe-free illumination to aim for near-perfect tiling in montage cryo-electron tomography.

B. Since detector-matching square beam significantly reduces beam overlapping, a major benefit out of this work would be potentially a high-resolution, 3-dimensional, and large field of view of, for example, cellular contexts.

We added the results as requested. Please see below.

Although the authors have shown tiled images at 0 and +/- 45 degrees, no montage tilt series is presented. In addition, the text at line 67 suggests that manual identification and alignment of overlapping areas was performed. How long does this take? I understand this work is a snapshot of an early development, but is there any thought of automated image tiling over the entire tilt range? I do not think anyone would like to manually tile images together.

We have introduced a modification in the already published package PACE-tomo that allows for data acquisition with a default distance of 1-camera Y length. With it, we have acquired an example 3x3 tilt-series on cellular samples to show the immediate benefit and application of this work as suggested by the reviewer.

Line 28-30: "Cryo-TEM imaging requires balancing the entire region of interest." This is true if we want to have both high resolution and large field of view. For other purposes like identifying holes of proper ice thickness and targets for data collection, montage are usually made at much lower magnifications. In these cases, we do not need high resolution and thus the balance. We reworded the sentence to be more precise. It now reads: "*High-resolution cryo-TEM typically comes at the cost of a reduced field of view; therefore, a balance between the pixel-size and the amount of context imaged around the region of interest is required.*"

Fig. 1C: Why do the square-beam images have much lower contrast than the round-beam counterpart? They are all FFI images except different beam shapes. The poor contrasts can also be seen in Supplemental Fig 2 at lines 264 and 268, respectively. Such differences do need explanations.

This is an effect of the normalization during the Jpeg conversion. The artifact is originated by the presence of a non-illuminated area that shows zero counts. We have corrected the figure and added the crops from the center to show a consistent contrast.

Fig 1D-E (1): The top and bottom edges of tile images have irregular shapes that change from tile to tile. If the irregular shapes correspond to the square C2 aperture, I do not understand why the shapes of tile images change from tile to tile. Please explain. Fig 1D-E (2): There are also visible gaps (white strips) both in the line and square montages. I do not understand why the authors allow the presence of the gaps in their montage tilt series. Are the authors going to reconstruct 3D volumes from the montage tilt series? If so, what is the strategy to fill the tiling gaps? If not, please explain how the montage tilt series is used.

There was a mistake in the cropping and stitching, which were performed manually using Fiji. Further, the data collection in the first version of the paper was performed in a semi-manual mode. Now, we can entirely rely on pre-calculated image shifts and can achieve near-perfect tiling. We have corrected the mistake and relied on the image shift calculations when preparing the montage for the new figures. This new procedure (now present in PACE-tomo provides continuity across tiles.

While square beam has a great potential to improve montage tilt series, the manuscript has not yet be convincing regarding how to efficiently and accurately tile images together and produce

high-resolution large field view from the tiled tilt series. I would like to read the revised manuscript should the authors be willing to address these questions.

In this manuscript, we demonstrated no impact on resolution when imaging with a square beam. An aspect associated with tiling and stitching still needs improvement, especially in subtomogram averaging, is how to deal with particles that may be present in different tiles across the tilt-series (this will occur across the tilt-series as the FOV changes along the Y-axis with the tilt). This aspect is linked to data analysis, and it is beyond the scope of this work.

Precise tiling relies essentially on software and tracking; we now show that it is possible to reconstruct tomograms with near-perfect tiling (See Figure 1 and Supplementary videos). We on-purpose did not optimize or correct the minor errors in image shift between the tiles to show the level of precision that can be achieved with this procedure as raw. Improved algorithms for data acquisition and tile alignment can attend to these errors.

During the revisions, we implemented a further procedure that can be used to align the tilt axis, the illumination profile, and the camera. The process is now fully explained in the supplementary material.

Decision Letter, first revision:

Dear Alex,

Thank you for submitting your revised manuscript "Square beams for optimal tiling in TEM" (NMETH-BC53378A). It has now been seen by the original referees and their comments are below. The reviewers find that the paper has improved in revision, and therefore we'll be happy in principle to publish it in Nature Methods, pending minor revisions to satisfy the referees' final requests and to comply with our editorial and formatting guidelines.

We are now performing detailed checks on your paper and will send you a checklist detailing our editorial and formatting requirements within two weeks or so. Please do not upload the final materials and make any revisions until you receive this additional information from us. Please provide a point-by-point rebuttal when you resubmit, for our convenience.

TRANSPARENT PEER REVIEW

Nature Methods offers a transparent peer review option for new original research manuscripts submitted from 17th February 2021. We encourage increased transparency in peer review by publishing the reviewer comments, author rebuttal letters and editorial decision letters if the authors agree. Such peer review material is made available as a supplementary peer review file. Please state in the cover letter 'I wish to participate in transparent peer review' if you want to opt in, or 'I do not wish to

participate in transparent peer review' if you don't. Failure to state your preference will result in delays in accepting your manuscript for publication.

ORCID

Sincerely,
Rita

Rita Strack, Ph.D.
Senior Editor
Nature Methods

Reviewer #1 (Remarks to the Author):

It appears that the square aperture exhibits a distortion range comparable to that of the circular aperture in the same microscope. The author has addressed the previous concerns in the review with revisions. This work is intriguing and will undoubtedly enhance tile/multi-tomogram data collection by maintaining consistent angles, thus significantly improving the data collection efficiency.

Reviewer #2 (Remarks to the Author):

The author have addressed the questions that reviewer asked in the revised manuscript and provided the supplemented tilt series and tomogram videos.

For the revised version, I have the following comments:

1. In Figure 1 D-I and supplementary videos, the author mentioned they are on purpose did not optimize the errors in image shift between tiles, I would recommend the authors show the optimized montage tomography data in main figure and use the current version of Figure 1D-I in supplementary material.
2. I can see the square boundaries of each tile in the montaged tilt series and tomograms, what's the main cause of these? Bad stitching or overexposure of the overlapped tile boundary. I did not see such boundaries in montaged tomography results from the papers of Peck, A. et. at and Yang, J. E. et. al. Since one of the important contribution of square beam will be benefit for the montage tomography, the authors should clarify its potential problems.
3. For supplementary figure S3, except for schematic graph, please also show the montaged tiles of cryoEM images at the specific tilt angles.

Author Rebuttal, first revision:

Response to the reviewers 2

We once again thank the reviewers for their time and effort in re-evaluating this work. Below we provide a point-to-point response to the comments in blue.

Reviewer #1:

Remarks to the Author:

It appears that the square aperture exhibits a distortion range comparable to that of the circular aperture in the same microscope. The author has addressed the previous concerns in the review with revisions. This work is intriguing and will undoubtedly enhance tile/multi-tomogram data collection by maintaining consistent angles, thus significantly improving the data collection efficiency.

We thank the reviewer for their feedback.

Reviewer #2:

Remarks to the Author:

The author have addressed the questions that reviewer asked in the revised manuscript and provided the supplemented tilt series and tomogram videos.

For the revised version, I have the following comments:

1. In Figure 1 D-I and supplementary videos, the author mentioned they are on purpose did not optimize the errors in image shift between tiles, I would recommend the authors show the optimized montage tomography data in main figure and use the current version of Figure 1D-I in supplementary material.

The beam-image shift matrices on our microscope have been calibrated, as can be seen by the accurate targeting of each tile resulting in the imaged features being aligned quite accurately between tiles in Figure 1D-F, even without further image processing to align the tiles. To make this point clearer, we have updated the figure legend from

“(F) High magnification crop of the joint between two tiles from panel E. In the upper section, imperfections in the stitching are visible, as no alignment or interpolation was performed when stitching. However, the apoferritin particles are clearly visible.”

to

“(F) High magnification crop of the joint between two tiles from panel E. In the upper section, despite no alignment or interpolation being performed when stitching, the image shifts are sufficiently accurate for providing contextual information. Apoferritin particles are clearly visible even at the stitching lines in the reconstructed tomograms.”

We further elaborate on this in our response to question 2.

2. I can see the square boundaries of each tile in the montaged tilt series and tomograms, what's the main cause of these? Bad stitching or overexposure of the overlapped tile boundary. I did not see such boundaries in montaged tomography results from the papers of Peck, A. et. at and Yang, J. E. et. al.

Since one of the important contributions of square beam will benefit the montage tomography, the authors should clarify its potential problems.

In the work by Peck et al. and Yang et al., the absence of stitching lines is because montaged tiles were collected with overlapping areas, which were then blended in a post-processing step to create a “seamless” montage. In this work, we chose to tile the square beams next to each other to show one possible data collection scheme that eliminates excessive sample damage. Because our tiles were collected without any overlap, even blending from one tile to the next emerges as an obvious stitching line. In principle, we do not expect stitching lines to be bad for downstream experiments such as sub-tomogram averaging or segmentation, as those experiments can be done on individual tiles, which can then be stitched back into the montage for visualization of the larger context.

We imagine that alternative data collection schemes may be implemented in the future, such as with an overlap between square tiles, which can then be blended as was done in Peck et al. and Yang et al. This will create a seamless montage. We added an experiment to demonstrate this possibility and have shown this now in a new Supplementary Figure S4, where the seams are minimally visible. The optimal data collection scheme depends on the purpose of the experiment, and a full treatment of possible collection and processing schemes is outside the scope of this paper.

We have elaborated on this in the text by including the following:

“While this data collection method eliminates sample overexposure, the lack of overlapping regions between tiles results in visible stitching lines (e.g., Figure 1D and G). We tested an alternate data collection scheme of a 3x3 montage on a carbon-foil apoferritin grid where each tile had a 5% overlap with its neighbor. The overlapping regions were then cross-correlated and blended, producing a nearly seamless montage (Supplementary Figure S4).”

3. For supplementary figure S3, except for schematic graph, please also show the montaged tiles of cryoEM images at the specific tilt angles.

As requested, we have included montaged tiles of cryo-EM images in Supplementary Figure S3.

Final Decision Letter:

18th Dec 2023

Dear Alex,

I am pleased to inform you that your Brief Communication, "Square beams for optimal tiling in TEM", has now been accepted for publication in Nature Methods. The received and accepted dates will be July 31, 2023 and Dec 18, 2023. This note is intended to let you know what to expect from us over the next month or so, and to let you know where to address any further questions.

Over the next few weeks, your paper will be copyedited to ensure that it conforms to Nature Methods style. Once your paper is typeset, you will receive an email with a link to choose the appropriate publishing options for your paper and our Author Services team will be in touch regarding any additional information that may be required.

You will receive a link to your electronic proof via email with a request to make any corrections within 48 hours. If, when you receive your proof, you cannot meet this deadline, please inform us at rjsproduction@springernature.com immediately.

Your paper will now be copyedited to ensure that it conforms to Nature Methods style. Once proofs are generated, they will be sent to you electronically and you will be asked to send a corrected version within 24 hours. It is extremely important that you let us know now whether you will be difficult to contact over the next month. If this is the case, we ask that you send us the contact information (email, phone and fax) of someone who will be able to check the proofs and deal with any last-minute problems.

If, when you receive your proof, you cannot meet the deadline, please inform us at rjsproduction@springernature.com immediately.

Once your manuscript is typeset and you have completed the appropriate grant of rights, you will receive a link to your electronic proof via email with a request to make any corrections within 48 hours. If, when you receive your proof, you cannot meet this deadline, please inform us at rjsproduction@springernature.com immediately.

Once your paper has been scheduled for online publication, the Nature press office will be in touch to confirm the details.

Content is published online weekly on Mondays and Thursdays, and the embargo is set at 16:00 London time (GMT)/11:00 am US Eastern time (EST) on the day of publication. If you need to know the exact publication date or when the news embargo will be lifted, please contact our press office after you have submitted your proof corrections. Now is the time to inform your Public Relations or Press Office about your paper, as they might be interested in promoting its publication. This will allow them time to prepare an accurate and satisfactory press release. Include your manuscript tracking number NMETH-BC53378B and the name of the journal, which they will need when they contact our office.

About one week before your paper is published online, we shall be distributing a press release to news organizations worldwide, which may include details of your work. We are happy for your institution or funding agency to prepare its own press release, but it must mention the embargo date and Nature Methods. Our Press Office will contact you closer to the time of publication, but if you or your Press Office have any inquiries in the meantime, please contact press@nature.com.

If you are active on Twitter, please e-mail me your and your coauthors' Twitter handles so that we may tag you when the paper is published.

Please note that *Nature Methods* is a Transformative Journal (TJ). Authors may publish their research with us through the traditional subscription access route or make their paper immediately open access through payment of an article-processing charge (APC). Authors will not be required to make a final decision about access to their article until it has been accepted. [Find out more about Transformative Journals](https://www.springernature.com/gp/open-research/transformative-journals)

To assist our authors in disseminating their research to the broader community, our SharedIt initiative provides you with a unique shareable link that will allow anyone (with or without a subscription) to read the published article. Recipients of the link with a subscription will also be able to download and print the PDF. As soon as your article is published, you will receive an automated email with your shareable link.

Please note that you and your coauthors may order reprints and single copies of the issue containing your article through Springer Nature Limited's reprint website, which is located at <http://www.nature.com/reprints/author-reprints.html>. If there are any questions about reprints please send an email to author-reprints@nature.com and someone will assist you.

Best wishes and happy holidays!

Rita

Rita Strack, Ph.D.
Senior Editor
Nature Methods